# Comparison of Glycemic Variability and Hypoglycemic Events in Hospitalized Older Adults Treated with Basal Insulin plus Vildagliptin and Basal–Bolus Insulin Regimen: A Prospective Randomized Study

**DOI:** 10.3390/jcm11102813

**Published:** 2022-05-16

**Authors:** Sol Batule, Analía Ramos, Alejandra Pérez-Montes de Oca, Natalia Fuentes, Santiago Martínez, Joan Raga, Xoel Pena, Cristina Tural, Pilar Muñoz, Berta Soldevila, Nuria Alonso, Guillermo Umpierrez, Manel Puig-Domingo

**Affiliations:** 1Servicio de Endocrinología y Nutrición, Hospital Germans Trias i Pujol, 08916 Badalona, Spain; sol_batule@hotmail.com (S.B.); aeramos.germanstrias@gencat.cat (A.R.); alec148@gmail.com (A.P.-M.d.O.); natfuente@hotmail.com (N.F.); santiagomartinezag@gmail.com (S.M.); soldevila.berta@gmail.com (B.S.); nalonso32416@yahoo.es (N.A.); 2Servicio de Medicina Interna, Hospital Germans Trias i Pujol, 08916 Badalona, Spain; jragaa.germanstrias@gencat.cat (J.R.); xpenape.germanstrias@gencat.cat (X.P.); ctural.germanstrias@gencat.cat (C.T.); mpmunozr.germanstrias@gencat.cat (P.M.); 3Department of Medicine, Emory University, Atlanta, GA 30322, USA; geumpie@emory.edu

**Keywords:** diabetes mellitus, vildagliptin, inpatient hyperglycaemia, older adults

## Abstract

Background: The basal–bolus insulin regimen is recommended in hospitalized patients with diabetes mellitus (DM), but has an increased risk of hypoglycemia. We aimed to compare dipeptidyl peptidase 4 inhibitors (DPP4-i) and basal–bolus insulin glycemic outcomes in hospitalized type 2 DM patients. Methods and patients: Our prospective randomized study included 102 elderly T2DM patients (82 ± 9 years, HbA1c 6.6% ± 1.9). Glycemic control: A variability coefficient assessed by continuous glucose monitoring (Free Style^®^ sensor), mean insulin dose and hypoglycemia rates obtained with the two treatments were analyzed. Results: No differences were found between groups in glycemic control (mean daily glycemia during the first 10 days: 152.6 ± 38.5 vs. 154.2 ± 26.3 mg/dL; *p* = 0.8). The total doses Kg/day were 0.40 vs. 0.20, respectively (*p* < 0.001). A lower number of hypoglycemic events (9% vs. 15%; *p* < 0.04) and lower glycemic coefficient of variation (22% vs. 28%; *p* < 0.0002) were observed in the basal–DPP4-i compared to the basal–bolus regimen group. Conclusions: Treatment of inpatient hyperglycemia with basal insulin plus DPP4-i is an effective and safe regimen in old subjects with T2DM, with a similar mean daily glucose concentration, but lower glycemic variability and fewer hypoglycemic episodes compared to the basal bolus insulin regimen.

## 1. Introduction

Hyperglycemia is a frequent finding in hospitalized patients (12.4–25%) [1]. Remarkably, more than 30% of patients with hyperglycemia detected during a hospitalization episode do not have a previous diagnosis of diabetes mellitus (DM) [2]. Hyperglycemia in hospitalized patients was consistently associated with a poor prognosis, especially in patients without a known history of diabetes [2]. A lack of adequate control of glycemia during hospitalization leads to a longer in-hospital stay, an increased incidence of infections and more hospital complications than patients without DM [1,3], which accounts for an increased necessity of healthcare resources in these patients [4,5].

Currently, different scientific societies recommend the administration of subcutaneous insulin as the treatment of choice for glycemic control in non-critical hospitalized patients using a basal–bolus regime [4,6,7]. Treatment with oral hypoglycemic agents in this context is discouraged due to theoretical limitations regarding their effectiveness and safety. Among them, a slow onset of action and, for some oral agents, a long duration of action were found, implying an insufficient flexibility to adapt to quickly changing requirements throughout the day and to other circumstances, such as fasting requirements associated with medical therapeutic and exploratory procedures [1,8]. In addition to these potential inconveniences, there are some limitations of efficacy and safety in hospitalized patients with type 2 DM (T2DM) regarding the number of randomized studies with oral drugs, which have conditioned its full implementation as a standard of care.

Several recent randomized trials demonstrated the potential effectiveness of dipeptidyl peptidase 4 inhibitors (DPP4-i) in specific groups of hospitalized patients. DPP4-i are well-tolerated oral drugs that demonstrated their efficacy and safety in association with other oral hypoglycemic agents and/or insulin [9]. DPP4-i do not cause hypoglycemia or weight gain, nor do they present significant drug interactions, which make them a very attractive therapeutic option for the treatment of T2DM, particularly in the elderly. Among this group of antidiabetic oral agents, there is currently only one safety study in subjects older than 75 years, performed with vildagliptin to support its use in elderly patients [10]. 

With the aim of simplifying and facilitating the implementation of an effective and safe treatment for hospital hyperglycemia, Umpierrez et al. recently published the results of an alternative to a basal–bolus regime for patients with T2DM admitted to non-critical units (conventional hospitalization). This regime is based on the administration of a daily dose of basal insulin and a single dose of the DPP4-i sitagliptin (Sita-Hospital study) [11]. The results of the study showed a non-inferiority efficacy in relation to a full regime of insulin. Similar findings were reported in a study of non-cardiac surgical and medical patients with T2DM, in which linagliptin was used [12,13].

Glycemic variability (GV) was proposed as a novel marker of glycemic control [14,15]. A large multicenter study concluded that GV was a much stronger predictor of intensive care unit (ICU) mortality than mean glucose concentrations [16]. The coefficient of variation (CV) was also demonstrated as a strong independent index for measuring GV as it corrects for mean glucose levels [17,18,19].

The aim of the present study was to compare the efficacy, safety and glycemic variability outcomes of a combined basal–DPP4-i regime compared to a conventional basal–bolus insulin regimen in internal medicine patients with T2DM.

## 2. Materials and Methods

A prospective randomized study was set up in a tertiary university Hospital Germans Trias i Pujol, in Badalona, Spain, in which a total of 102 patients were included (Figure 1). 

Patients with T2DM aged ≥65 years old, for whom all of the following concurrent conditions were present, were included in the study: (1) plasma glycemia on admission at less than 400 mg/dL and (2) treatment at home with any combination of oral antidiabetic drugs or with insulin therapy at a daily dose less 0.6 IU/Kg/day. Exclusion criteria were any of the following: (1) HbA1c >9%, (2) glycaemia at admission ≥400 mg/dL, (3) treatment with glucocorticoids (dose >5 mg/day of prednisone or equivalent), (4) previous treatment with insulin at a total daily dose ≥0.6 IU/Kg, and (5) any of the following active clinical situations or previous antecedents: acute myocardial infarction, acute pancreatitis, diagnosis of type 1 diabetes mellitus and/or hepatic cirrhosis. In all of the latter, the usual basal–bolus regime was administered, and the patients were not included in the study.

All participants provided written informed consent before the start of the protocol. Patients were randomly assigned to one of two treatment regimens according to the side of the room they were hospitalized in (hallway A or hallway B). The dose of insulin was started in both groups based on previous treatment of DM and capillary blood glucose (CBG) concentration at admission as indicated in Supplementary Appendix A. In the basal–bolus group, half of the insulin dose was prescribed as basal insulin (glargine) once daily at bedtime and half as rapid-acting insulin (aspart) divided into three equal doses before meals. For patients in the basal–DPP4-i group, the complete calculated dose of insulin was administered as long-acting analogue once daily at bedtime, and vildagliptin dose was calculated according to GFR: 100 mg/day if GFR ≥50 mL/min per 1.73 m^2^ and 50 mg/day if GFR <50 mL/min per 1.73 m^2^. 

The goal of the treatment was to maintain glycemic control between 140 and 180 mg/dL. In both groups, CBG was measured before meals and bedtime. In the case of hypoglycemia (defined as grade 1 between 56 and 70 mg/dL, grade 2 < 56 mg/dL and grade 3 < 56 mg/dL plus neuroglycopenia symptoms), insulin treatment was reduced as indicated in Appendix A. If CBG was >300 mg/dL, a correction dose of rapid-acting insulin was administered, as also indicated in Appendix A. Failure of treatment in the basal–DPP4-i group was defined as CBG > 300 mg/dL in two consecutive measures even after dose upload correction, and in these cases, the patient was switched to a basal–bolus regime. 

In a subset of patients (*n* = 20), a FreeStyle^®^ glycemic sensor was used to assess glycemic variability. Hypoglycemia detected by the sensor was confirmed by a CBG measurement, which was used for therapeutic decisions. The percentages of time in the various glycemic ranges were assessed according to the International Consensus on Time in Range [20] for older/high risk T2DM patients. Hyperglycemia was defined as: grade 1 (>50% of time between 181 and 250 mg/dL) and grade 2 (>10% of time >250 mg/dL). Hypoglycemia in this group was defined as a glycemic level below 70 mg/dL >1% of time. Acceptable time (time in range) in range was defined as >50% of time between 70 and 180 mg/dL.

HbA1c was measured within 24 h of randomization if the patient did not have a determination in the last 3 months. The degree of glycemic control, glycemic CV, mean insulin dose and hypoglycemia rates observed with the two therapeutic modalities were used for statistical analyses. 

For the statistical analysis purpose, the first ten days of hospital stay were considered. Continuous variables are presented as mean ± standard deviations (SD). Comparisons between both therapeutic strategies were made using the Student’s test for independent samples, the Fisher test or χ^2^ (categorical variables). All the statistical analyses were performed with IBM SPSS Statistic software version 26 (IBM Corporation, New York, NY, USA).

## 3. Results

A total of 102 patients were eligible for the study. Of these, eight patients were excluded (four of them due to early discharge, two due to glucocorticoid treatment initiation after recruitment and two due to the failure of treatment). We analyzed 94 patients, 50 of which were included in the basal–bolus regime and 44 in the basal–DPP4-i regime (Figure 1).

Among all patients, the main reasons for admission were heart failure (30.8%), followed by respiratory infections (20.2%) and non-respiratory infections (16%). The causes for other patients were mostly part of geriatric syndrome and included consumptive syndrome, renal failure, confusional syndrome and falls, although none of them were significantly different regarding their frequency in both treatment arms. Those in the basal–bolus group compared to those in the basal–DPP4-i group, showed significant differences in sex (female 56% vs. 34%, *p* = 0.04, respectively) and weight (71.8 ± 16 Kg vs. 81.3 ± 18 Kg, *p* = 0.008, respectively). No significant differences were observed regarding HbA1c (6.7 ± 1.2% vs. 6.6 ± 0.9%) or any other baseline variable such as age, admission blood glucose, previous outpatient antidiabetic treatment, length of hospital stay and biochemical parameters (Table 1).

There were no statistical differences o → n any day of mean blood glucose measurements during the study period between the basal–bolus and basal–vildagliptin groups (157 ± 36.9 mg/dL vs. 145 ± 29.5 mg/dL, *p* = 0.103, respectively) (Figure 2). As expected, the mean basal insulin dose requirements were significantly higher for the basal–bolus group, and a lower number of grade 1 hypoglycemia was observed in the basal–DPP4-i group. Regarding CV, the mean was <36% in both groups, being statistically lower in the basal–vildagliptin group (Table 2).

In patients who used the FreeStyle^®^ glycemic sensor, when comparing the basal–bolus group with the basal–vildagliptin group, we found no statistical differences in glycemic parameters, with a mean interstitial glycemia of 157 ± 43.6 mg/dL vs. 132 ± 23.8 mg/dL (*p* = 0.133), time in range 74.3 ± 18.8% vs. 82.3 ± 17.7% (*p* = 0.341), and CV 29.2 ± 6.8% vs. 23.7 ± 3.8% (*p* = 0.135). In both groups, there was no grade 1 hyperglycemia. Grade 2 hyperglycemia was found in two patients in the basal–bolus group and one patient in the basal–vildagliptin group. Hypoglycemia events were present in four patients in the basal–bolus group and in two patients in the basal–vildagliptin group.

## 4. Discussion

The present study confirms the feasibility of implementing an easier, safer and equally effective treatment modality using basal insulin plus DPP4-i, compared to the classic basal–bolus insulin regimen for the management of hyperglycemia in hospitalized older adults. In addition to demonstrating that good glycemic control can be achieved with both treatment modalities with a mean glycemic value of about 150 mg/dL, our study indicates that the number of hypoglycemic episodes is lower, less intense and particularly fewer in absolute terms with a basal–DPP4-i combination compared to a basal–bolus regimen. Moreover, the glycemic variability was lower with basal–DPP4-i treatment.

Hypoglycemia has serious consequences in terms of hospital outcomes, in-hospital length of stay, and in general, it increases the health resources consumed in one hospital episode stay. Hypoglycemia was associated with cardiovascular events, myocardial infarction and stroke due to the impact of the sharp increase in circulating catecholamines induced by the decrease in glycemic circulating levels [21]. In addition, it has recently been confirmed that glycemic variability is a very important treatment target in every diabetic patient. Consistent data indicate that glycemic variability is associated with increased oxidative stress [22] and other deleterious biologic processes that affect general health, particularly in older adults admitted with cardiovascular or infectious episodes.

In addition to efficacy and safety issues, this alternative regime using DPP4-i is more convenient for the patient and the nursing staff who provide care in hospital conventional beds. The decrease in insulin injections per se, as well as the advantage of the non-activity of DPP4-i molecules at low–normal glycemic levels, provides comfortability and safety in old and frail patients who may be unable to complete their intake of a given hospital meal due to increased anorexia or inappetence for hospital food.

The present study has some weaknesses, including a relative low number of subjects and a small subsample of patients using continuous glucose monitoring. In addition, the group of previously studied patients had an overall good control according to the HbA1c value at hospital entry. However, as for the potential strengths of the present study, the prototype of the included patients was typical of those mostly admitted in the hospital. In this regard, it seems logical that patients with relatively well-controlled DM did not require heavy dosing with a full basal–bolus insulin regime, which in fact may be more harmful than a single dose of long-acting insulin plus a DPP4-i for covering the prandial glycemia in a patient with an otherwise reasonable beta cell reserve. Additionally, the single-center nature of our study warranted the homogeneity of the procedures and patient inclusion.

## 5. Conclusions

The basal–bolus insulin therapy regimen remains a useful treatment for many inpatients, especially those with symptomatic hyperglycemia, poor glycemic control prior to admission, and those who fail to maintain glucose control with basal insulin, plus DPP4-i. However, the high rate of hypoglycemia represents a major limitation, and the active daily review of insulin dosage is mandatory, which is quite time consuming and requires certain expertise. Our results indicate that an alternative regime with the combination of basal insulin plus DPP4-I, was effective and safer than a basal–bolus regime as less hypoglycemic episodes were detected and with the added value of reduced glycemic variability in older well-controlled diabetic patients. Moreover, improved convenience for patients and the nursing staff may contribute to recommending this kind of treatment modality as the standard of care for controlling inpatient glycemia in most of adults with type 2 diabetes.

## Figures and Tables

**Figure 1 jcm-11-02813-f001:**
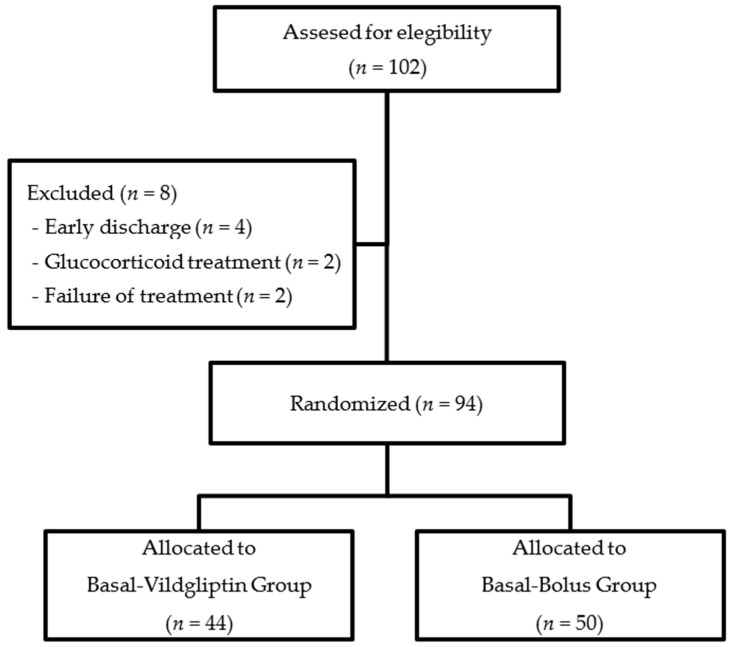
Flowchart of study population.

**Figure 2 jcm-11-02813-f002:**
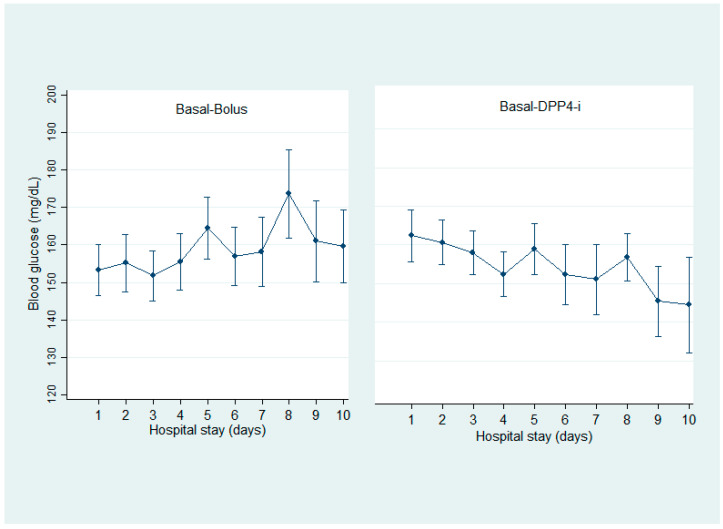
Mean daily blood glucose concentrations during hospital stays. Values are shown as mean ± standard deviations.

**Table 1 jcm-11-02813-t001:** Baseline characteristics of all the patients.

Variable	Basal–Bolus (*n* = 50)	Basal–DPP4-i (*n* = 44)	*p*-Value
Age (years)	78.2 ± 15	80.5 ± 7	0.35
Gender			
Male, *n* (%)	22 (44)	29 (66)	0.04 *
Female, *n* (%)	28 (56)	15 (34)	
Weight	71.8 ± 16	81.3 ± 18	0.008 *
BMI (kg/m^2^)	28.53 ± 5.8	30.7 ± 6	0.05
Duration of diabetes (years)	15.4 ± 6	13.6 ± 6	0.13
Admission diabetes therapy			0.91
1 OAD, *n* (%)	18 (36)	19 (43.2)	
2 or more OAD, *n* (%)	7 (14)	12 (27.3)	
OAD + basal insulin, *n* (%)	11 (22)	8 (18.2)	
Basal insulin, *n* (%)	9 (18)	2 (4.5)	
Basal–bolus ± OAD	4 (8)	2 (4.5)	
Admission blood glucose (mg/dL)	171.9 ± 69	181.5 ± 68	0.5
HbA1c (%)	6.7% ± 1.2	6.6 ± 0.9	0.85
GFR (mL/min/1.73 m^2^)	48.8 ± 24	48.1 ± 25	0.87
Length of hospital stay (days)	11.4 ± 9.3	11.9 ± 10	0.78

Abbreviations: OAD, oral antidiabetic agent; HbA1c, glycated hemoglobin, GFR, glomerular filtration rate. * Differences between groups <0.05. Data are mean ± standard deviation.

**Table 2 jcm-11-02813-t002:** Variables associated with glycemic control.

Variable	Basal–Bolus (*n* = 40)	Basal–Vildagliptin (*n* = 34)	*p*-Value
Mean CBG	157 ± 36.9	145 ± 29.5	0.103
Insulin dose			
Total mean insulin dose, IU/day	29.1 ± 11.8	15.3 ± 5.1	0.001 *
Total mean insulin dose, IU/kg/day	0.4 ± 0.17	0.2 ± 0.1	<0.001 *
Total glargine insulin dose, IU/day	14.1 ± 6	15.4 ± 5.1	0.105
Total aspart insulin dose, IU/day	14.4 ± 7.6	-	-
Hypoglycaemic events			
Patients with any blood glucose < 70 mg/dL, *n* (%)	8 (20)	1 (3.4)	0.023 *
Patients with any blood glucose < 54 mg/dL, *n* (%)	1 (2.5)	-	-
Hyperglycaemic events			
BG > 180 mg/dL, *n* (%)	10 (25)	5 (14.7)	0.386
CV (%)	28	22	<0.001 *

Abbreviations: CBG, capillary blood glucose; BG, blood glucose; CV, coefficient of variation. * Differences between groups <0.05. Data are mean ± standard deviation.

## Data Availability

The data are available from the corresponding author on request.

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
