# Peer review of "Comparison of Glycemic Variability and Hypoglycemic Events in Hospitalized Older Adults Treated with Basal Insulin plus Vildagliptin and Basal–Bolus Insulin Regimen: A Prospective Randomized Study"

_jcm, 2022, doi:10.3390/jcm11102813_

Round 1

Reviewer 1 Report

The manuscript entitled “Comparison of glycemic variability and hypoglycemic events in hospitalized older adults treated with basal insulin plus vildagliptin and basal-bolus insulin regimen: A prospective randomized study” evaluated the efficacy and safety of two treatments for in-hospital blood glucose control in a small cohort of elderly people. Basal-DPP4i treatment was associated with hypoglycaemia less often when compared to basal-bolus treatment, as well as a lower coefficient of variation.   

Minor comments 

  • Why were patients not allocated to basal-vidagliptin and basal-bolus in a manner 1:1 rather than 44:50?
  • The authors reported the causes of hospitalization in 67% of the patients. What was the remaining cause of hospitalization in 33% of patients? Was there a difference in the causes of hospitalization between the groups?
  • Abstract section: Review the standard deviation of HbA1c (6.6%±19). Is it 1.9% instead of 19%?
  • Were the episodes of hypoglycaemia symptomatic or asymptomatic?

Reviewer 2 Report

Dear Authors,

  1. In my opinion conlsusions should be written as separate point and this point should be more extensive.
  2. In my opinion should be a part of test with inclusion critieria and their description

Round 2

Reviewer 2 Report

Dear Authors ,

  1. In my opinion text with inclusion and  exclusion critieria are incomplete and unreadable.
